# Sex Disparities and Female Reproductive and Hormonal Factors Associated with Risk of Pancreatic Cancer in the European Prospective Investigation into Cancer and Nutrition (EPIC) Cohort

**DOI:** 10.3390/cancers17142275

**Published:** 2025-07-08

**Authors:** Verena A. Katzke, Srimanti Dutta, Anna Rasokat, Livia Archibugi, Gabriele Capurso, Giulia Peduzzi, Manuel Gentiluomo, Federico Canzian, Anne Kirstine Eriksen, Anne Tjønneland, Christina C. Dahm, Therese Truong, Marianne Canonico, Nasser Laouali, Matthias B. Schulze, Rosario Tumino, Giovanna Masala, Claudia Agnoli, Lucia Dansero, Salvatore Panico, Marta Crous-Bou, Esther Molina-Montes, Ane Dorronsoro, María-Dolores Chirlaque, Marcela Guevara, Salma Tunå Butt, Malin Sund, Sofia Christakoudi, Elom K. Aglago, Elisabete Weiderpass, Marc Gunter, Daniele Campa, Rudolf Kaaks

**Affiliations:** 1Division of Cancer Epidemiology, German Cancer Research Center (DKFZ), 69120 Heidelberg, Germany; srimanti.dutta@dkfz.de (S.D.); r.kaaks@dkfz.de (R.K.); 2Clinic I for Internal Medicine, University Clinic Cologne, 50937 Cologne, Germany; anna.rasokat@uk-koeln.de; 3Pancreato-Biliary Endoscopy and Endosonography Division, Pancreas Translational and Clinical Research Center, IRCCS San Raffaele Hospital, 20129 Milan, Italy; archibugi.livia@hsr.it (L.A.); capurso.gabriele@hsr.it (G.C.); 4Department of Biology, University of Pisa, 56126 Pisa, Italy; giulia.peduzzi@phd.unipi.it (G.P.); manuel.gentiluomo@unipi.it (M.G.); daniele.campa@unipi.it (D.C.); 5Division of Genomic Epidemiology, German Cancer Research Center (DKFZ), 69120 Heidelberg, Germany; f.canzian@dkfz-heidelberg.de; 6Danish Cancer Society Research Center, 2100 Copenhagen, Denmark; ake@cancer.dk (A.K.E.); annet@cancer.dk (A.T.); 7Department of Public Health, University of Copenhagen, 1165 Copenhagen, Denmark; 8Department of Public Health, Aarhus University, 8000 Aarhus, Denmark; ccd@ph.au.dk; 9UVSQ, Inserm “Exposome and Heredity” Team, Université Paris-Saclay, CESP U1018, Gustave Roussy, 91990 Villejuif, France; therese.truong@inserm.fr (T.T.); marianne.canonico@inserm.fr (M.C.); nasser.laouali@inserm.fr (N.L.); 10Deutsches Institut für Ernährungsforschung Abteilung Molekulare Epidemiologie Arthur-Scheunert-Allee 114–116, 14558 Nuthetal, Germany; mschulze@dife.de; 11Hyblean Association Epidemiology Research, AIRE ONLUS Ragusa, 97100 Ragusa, Italy; rosario.tumino@asp.sr.it; 12Institute for Cancer Research, Prevention and Clinical Network (ISPRO), 50139 Florence, Italy; g.masala@ispro.toscana.it; 13Epidemiology and Prevention Unit, Department of Research, Fondazione IRCCS Istituto Nazionale dei Tumori, Via Venezian, 1, 20133 Milan, Italy; claudia.agnoli@istitutotumori.mi.it; 14Centre for Biostatistics, Epidemiology, and Public Health (C-BEPH), Department of Clinical and Biological Sciences, University of Turin, 10043 Orbassano, Italy; lucia.dansero@unito.it; 15Dipartimento di Medicina Clinica e Chirurgia, Federico II University, 80138 Naples, Italy; salvatore.panico@unina.it; 16Unit of Nutrition and Cancer, Cancer Epidemiology Research Program, Catalan Institute of Oncology (ICO)—Bellvitge Biomedical Research Institute (IDIBELL), L’Hospitalet de Llobregat, 08908 Barcelona, Spain; marta.crous@iconcologia.net; 17Department of Epidemiology, Harvard T.H. Chan School of Public Health, Boston, MA 02115, USA; 18Department of Nutrition and Food Science, Campus of Cartuja, University of Granada, 18071 Granada, Spain; memolina@ugr.es; 19CIBER of Epidemiology and Public Health (CIBERESP), 28029 Madrid, Spain; mdolores.chirlaque@carm.es; 20Instituto de Investigación Biosanitaria ibs.GRANADA, 18012 Granada, Spain; 21Institute of Nutrition and Food Technology (INYTA) ‘José Mataix’, Biomedical Research Centre, University of Granada, 18071 Granada, Spain; 22Ministry of Health of the Basque Government, Sub Directorate for Public Health and Addictions of Gipuzkoa, 20013 San Sebastian, Spain; a-dorronsoroerauskin@euskadi.eus; 23Biodonostia Health Research Institute, Epidemiology of Chronic and Communicable Diseases Group, 20014 San Sebastián, Spain; 24Department of Epidemiology, Regional Health Council, IMIB-Arrixaca, Murcia University, 3003 Murcia, Spain; 25Instituto de Salud Pública y Laboral de Navarra, 31003 Pamplona, Spain; mguevare@navarra.es; 26Centro de Investigación Biomédica en Red de Epidemiología y Salud Pública (CIBERESP), 28029 Madrid, Spain; 27Navarra Institute for Health Research (IdiSNA), 31008 Pamplona, Spain; 28Department of Surgery, Institution of Clinical Sciences, Skåne University Hospital, Lund University, 21428 Malmö, Sweden; salma.butt@med.lu.se; 29Department of Surgical and Perioperative Sciences, Umeå University, 90187 Umeå, Sweden; malin.sund@umu.se; 30Department of Surgery/CLINICUM, University of Helsinki, 00100 Helsinki, Finland; 31Department of Epidemiology and Biostatistics, School of Public Health, Imperial College London, St Mary’s Campus, Norfolk Place, London W2 1PG, UK; s.christakoudi@imperial.ac.uk (S.C.); kouassivi-elom.aglago@enovalife.com (E.K.A.); 32Department of Inflammation Biology, School of Immunology and Microbial Sciences, King’s College London, London WC2R 2LS, UK; 33International Agency for Research on Cancer—WHO, 69008 Lyon, France; weiderpasse@iarc.who.int; 34Nutrition and Metabolism Branch, International Agency for Research on Cancer—WHO, 69009 Lyon, France; gunterm@iarc.who.int

**Keywords:** pancreatic cancer, sex differences, reproductive factors, hormonal factors, women, EPIC study

## Abstract

Men are more often diagnosed with pancreatic cancer than women. With this research, we aim to increase understanding of the reasons for this sex-related difference, which might be due to different risk profiles in men and women, or hormonal and reproductive factors in women might play an important protective role. In a large prospective cohort with almost 1300 pancreatic cancer cases, we found that men indeed have a higher risk of developing pancreatic cancer. Risks of overweight, diabetes, alcohol consumption and smoking did not differ by sex. However, the longer a woman breastfeeds her children and the longer she takes hormones during menopause, the lower her risk of pancreatic cancer. Further research is necessary to understand the biological mechanisms and also what type of hormone likely plays a fundamental role.

## 1. Introduction

Pancreatic cancer (PC) is the fifth leading cause of cancer-related mortality in the WHO Europe region, accounting for 7.1% of cancer deaths [1]. Although several aetiologicaletiological and morphological types of PC exist, pancreatic ductal adenocarcinoma (PDAC) accounts for more than 90% of PC [2]. Established epidemiological risk factors are tobacco smoking, body fatness, long-standing diabetes mellitus type II, chronic pancreatitis and family history of PC, whereas the evidence for heavy alcohol drinking and consumption of red and processed meat is limited but suggestive [3,4,5]. Genetics also plays an important role in PDAC predisposition, with rare, low-frequency, and common risk variants identified [6,7]. Furthermore, worldwide age-standardised incidence and mortality rates of PC are higher in men than in women [1,8], although the size of this sex difference varies across countries [9]. In previous analyses, the higher incidence rates among men could not be fully explained by differences in established risk factors [8,9], which led to the hypothesis that female-specific hormonal or reproductive factors might exert a protective effect against PC. Preclinical evidence on the existence and functioning of oestrogen receptors in pancreatic carcinogenesis renders this hypothesis biologically plausible [10,11,12,13], and several cohort [14,15,16] and case–control studies [17,18] have suggested a protective role for female hormones against PC development.

However, epidemiological studies on the effect of reproductive/hormonal factors on pancreatic carcinogenesis produced heterogeneous results, and analyses of larger (prospective) cohort studies to investigate hormonal and reproductive predictors of PC have been called for [19,20,21,22,23]. An earlier investigation in the European Prospective Investigation into Cancer and Nutrition (EPIC) cohort examined reproductive predictors for PC, but with a small number of incident PC cases (No. = 304) [24]. Sex disparities in incidence rates, heterogeneity of previous investigations, and the obvious need for larger cohort studies call for this updated research within the EPIC cohort, a well-defined prospective cohort study with 717 female and 577 male incident cases of PC. Our aim was to investigate associations of established risk factors for pancreatic cancer by sex, followed by in-depth analyses of hormonal and reproductive factors and their associations with pancreatic cancer risk in women, including parity, breastfeeding, and use of hormones.

## 2. Materials and Methods

### 2.1. Study Design

The EPIC cohort is an ongoing, multicentre European cohort study, comprising around 370,000 women and 150,000 men. The design, rationale and objectives of the EPIC study have been described in detail elsewhere [25,26]. In brief, participants in the age range of 35–70 years (40–70 for men) were recruited between 1992 and 2000 in 23 centers across 10 European countries to investigate aetiological associations of lifestyle, diet, and environmental factors with risk of common chronic diseases, including cancer. The study centers were recruited mainly from the general adult population, but also from health-insured, health-conscious populations, or blood donors. In France, Norway, Utrecht, and Naples, only women were recruited. Participant eligibility within each cohort was based on geographic or administrative boundaries and on a predefined age range. At baseline, all study participants provided comprehensive questionnaire data, including smoking history, alcohol consumption, previous history of surgeries (e.g., hysterectomy), illnesses (e.g., diabetes), level of schooling (as a proxy for socio-economic status), reproductive history, and hormone use. Duration of menstrual cycles, number of full-term pregnancies, age at first full-term pregnancy, duration of breastfeeding, and menopausal status were derived from a combination of several variables (see Section A.1. “Detailed Information on creation of reproductive and hormonal factors in EPIC”). In addition, anthropometric measures (body height, body weight, waist, and hip circumference) were taken, and blood was drawn from about 80% of the participants. Baseline characteristics of participants were not reassessed across all centers during follow-up of the study.

### 2.2. Follow-Up for Cancer Incidence and Vital Status, PC Characteristics

Cancer incident cases and deaths in EPIC are mainly recorded by linkage of the cohort data to cancer and death registries, with the exception of France, Germany and Greece, where a combination of methods was applied (active follow-up of study participants or their next-of-kin, retrieval of clinical records, and/or cancer registry linkages). Our study includes all incident cases of exocrine pancreatic cancer with ICD codes C25 (25.0–25.3, 25.7–25.9). Incident occurrences of other malignant tumours preceding the diagnosis of pancreatic cancer, except for non-melanoma skin cancer, were censored at the date of diagnosis, resulting in 717 female and 577 male incident pancreatic cancer cases (Figure 1).

Censoring dates varied by country, from 2009 to 2014, with a median follow-up time for PC of 10 years (IQR 6–13 years). Seventy percent of PC No. = 914) were microscopically confirmed (histological and/or microscopical examination or autopsy), whereas diagnosis of 374 cases (30%) was based only on clinical examination or imaging. Fourty-one percent (No. = 534) of tumours were reported to be situated in the head of the pancreas, ten percent (123) in the pancreas’ body, and eight percent (106) in the tail, while no tumour sub-localisation was specified for 475 cases (37%). Due to the relatively large proportions of missing data for tumour sub-localisation (37%), staging (68%), and grading (84%), inferential analyses were not stratified by these factors. Limited statistical power did not allow for analyses restricted to microspically confirmed cases. Above mentioned frequencies did not differ by sex.

Due to recent legal issues, the Greek component of the EPIC study had been excluded from all projects; the Norwegian study participants were withdrawn from this project due to issues related to GDPR (general data protection regulation).

### 2.3. Statistical Analyses

Cox proportional hazards models were applied to investigate associations of age (continuous), diabetes (yes, no), BMI (by WHO categories and continuous), body height (continuous), schooling (categories), smoking habits, alcohol consumption (in standard glasses categories and continuous), and meat intake (by quartiles and continuous) with pancreatic cancer risk overall and by sex. For smoking habits, four distinct exposures were investigated: smoking status (never, former, current), cigarettes smoked per day (only among current and former smokers, in ½ pack categories and continuous), duration of smoking (only among current and former smokers, in 10 years categories and continuous), and time since quitting smoking (only among former smokers, in 10 years categories and continuous).

In women only, Cox proportional hazards models were fitted separately for each reproductive and hormonal factor to assess their associations with PC risk in women. Predictors were age at menarche (in years), cumulative duration of menstrual cycles (in years) with and without use of oral contraceptives (OCs), full-term pregnancies (FTPs, yes/no), number of FTPs, age at first FTP (in years), breastfeeding (yes/no), cumulative duration of breastfeeding (in months), menopausal status (pre, peri/unknown, post), age at menopause (in years), every use of OC (yes/no), cumulative duration of OC use (in years), every use of hormone replacement therapy (HRT, yes/no), cumulative duration of HRT use among HRT users (in years), hysterectomy (yes/no), and ovariectomy (yes/no).

Age constitutes the underlying time scale in Cox models such that age at recruitment and age at censoring define entry and exit times, respectively. The models were stratified by study centre to avoid bias due to differences in recruitment methods and outcome ascertainment between centres. Models were run crude and adjusted for known risk factors of PC [4,27,28], namely age (indirectly by using age as the underlying time scale), smoking history (never/former quitting ≤10, 11–20, 20+ years/current smoking 1–15, 16–25, 26+ cigarettes per day, unknown), daily alcohol consumption at recruitment (g/day, continuous), Meat intake (g/d, continuous) history of type-2 diabetes (no, yes, unknown), body mass index (BMI, kg/m^2^, continuous), education (level of schooling) and body height (cm, continuous). Information on pancreatitis and family history of PC is not available in EPIC. Additional adjustment for meat intake at baseline, age at recruitment (in 5-year categories) or birth cohort (in 10-year categories) had no effect on Hazard Ratios (HR) or 95% Confidence Intervals (CI) and were, therefore, omitted from all models. Interaction terms for pregnancies and breastfeeding, as well as menopausal status and HRT use, proved to be non-significant and were, therefore, not included in the models. In exploratory analyses, all models were re-run by menopausal status (pre, peri/unknown, post).

Underlying assumptions of Cox proportional hazards models were checked: (a) predictor variables are assumed to contribute to the model in a linear way on the log scale, (b) the proportional hazard (PH) assumption was assessed by testing for a zero slope in the regression of the scaled Schoenfeld residuals, (c) non-informative censoring: all participants considered in this study were followed from the date of enrolment in EPIC to the date of PC diagnosis, the date of death, or the date of the last completed follow-up, whichever occurred first.

SAS 9.4 was used for creating the dataset and for data preparation. Statistical analyses were performed in SAS 9.4 (Statistical Analysis System, RRID:SCR_008567) and R version 4.4.0 (R Project for Statistical Computing, RRID:SCR_001905). All statistical tests are two-sided with significance level α = 0.05.

## 3. Results

The median ages at PC diagnosis were 68 and 67, for women and men, respectively (Table 1). In women, PC cases had a slightly higher median BMI than the non-cases (25.0 vs. 24.0). There was a higher proportion of current smokers (27% vs. 18% for women, 38% vs. 28% for men) and of individuals with a history of type 2 diabetes (4% vs. 2% for women, 6% vs. 3% for men) amongst incident cases of PC than amongst non-cases. Men were more likely to drink more alcohol per day, consumed more meat, were more often smokers and smoked more cigarettes than women.

Those who developed PC, and did not use OC, tended to have had more years of menstrual cycles (median 32 vs. 29 years), breastfed more often (70% vs. 63%) and were mostly postmenopausal at recruitment (73% vs. 43%), which was expected given the median age of onset of PC, compared to the non-cases. The proportion of HRT users was higher amongst PC incident cases (30% vs. 25%), whereas the use of OC was lower (47% vs. 59%). The latter flipped considering only women postmenopausal at recruitment (33% vs. 36%, not in tables).

Cox proportional hazards models showed a 1.31-fold higher risk of PC for men compared to women (HR, 95% CI 1.24–1.57) after adjustments for age, smoking history, BMI, diabetes, and alcohol consumption, stratifying for study center (Table 2). For both sexes, age, diabetes, current smoking, cigarettes smoked per day, duration of smoking and times since quitting were significantly associated with PC risk. Only among women, body height and meat intake at recruitment were significantly associated with PC risk (HR 1.12, 95% CI 1.05–1.19, per 5 cm and HR 1.18, 95%CI 1.02–1.36, per 100 g/d, respectively). Further, HRs for current smoking were slightly higher in men than in women (HR 1.83 vs. 1.74) but number of cigarettes smoked per day were higher in women than in men (HR 1.42 vs. 1.21, per 10 cigarettes/day). Stratifying by menopausal status, diabetes, greater body height, and greater alcohol consumption were associated with PC risk only in postmenopausal but not in premenopausal women (Appendix A Table A1). Premenopausal women who smoked at baseline, smoked more cigarettes per day and experienced longer smoking durations were at greater risk than postmenopausal women with the same risk factor profile.

Table 3 gives HRs with 95% CIs for all reproductive factors (a) adjusted for age and stratified by study centre and (b) fully adjusted for (a) and BMI, body height, daily alcohol consumption, smoking history, type 2 diabetes status, and level of schooling. We observed a slight, albeit not significant, tendency towards inverse associations with PC for age at menarche, number of FTPs, age at first FTP, and surgeries, and direct but non-significant associations for OC use, cumulative duration of OC use, and cumulative duration of menstrual cycles without OC use. Statistically significant, cumulative duration of HRT use of more than 2.4 years (median) was inversely associated with risk of PC (HR 0.72, 95% CI 0.54–0.97). Further, duration of breastfeeding was associated with a lower risk of PC: Compared with those who had breast-fed for 5.7 months or less, women with a cumulative time of breast-feeding of more than 5.7 months had a 23% lower risk (median, fully adjusted HR 0.77, 95% CI 0.64–0.93).

In perimenopausal women (N cases = 102, predominantly 46–54 years old), being 24 years of age or older at first FTP was associated with a reduced risk of PC (HR 0.58, 95% CI 0.35–0.97) compared to those younger than 24 years. Cumulative duration of breast feeding, as observed among all women, was inversely associated with PC risk also among postmenopausal women (N cases = 524, predominantly ≥55 years of age, HR 0.80, 95% CI 0.65–0.99, with >6 months of feeding compared to less). Further, cumulative duration of HRT use was inversely associated with PC risk (HR 0.66, 95% CI 0.47–0.93, >3 versus ≤3 years, median) among postmenopausal women only (Appendix A Table A2).

## 4. Discussion

Research into possible associations between female reproductive and hormonal factors and PC risk has been prompted by the lower incidence of PC in women compared to men. This risk difference between sexes was confirmed in the EPIC study, which showed that men had a 1.3-times greater risk of developing PC relative to women after multivariable adjustment. Interestingly, as for epidemiological risk factors, greater attained body height and increased meat consumption were associated with PC risk only in women. In line with previous research, the present study does not provide straightforward evidence supporting the hypothesised protective effect of female sex hormones. However, we observed statistically significant inverse associations between the cumulative duration of breastfeeding and HRT use and PC risk after full adjustment for smoking history, BMI, diabetes, alcohol consumption, and level of schooling among women. Other reproductive and hormonal factors were not significantly associated with PC risk. A previous investigation within EPIC on 304 incident PC cases conducted by Duell and colleagues found only younger age at menarche (<12 years) to be associated with risk of PC [24], a finding which we did not confirm.

In our study, women smoke less than men, a common observation across Western populations [29,30]. Nevertheless, in our cohort, women’s risk of PC is slightly lower to that of men if they smoked at recruitment (HR 1.74 vs. 1.83), but potentially greater the more cigarettes they smoked per day (HR 1.42 vs. 1.21). Findings on tobacco consumption by other cohorts are controversial, with some showing stronger risks in women whereas others have not observed **sex disparities** [31,32,33,34]. Our novel finding of a direct association between adult attained body height and PC risk in women has not been observed by others so far. A meta-analysis of cohort studies found a weak positive association between body height and PC risk, which was of similar size among men and women, although only statistically significant among men [35]. It is important to note that adult attained body height is a marker for genetic, environmental, hormonal, and nutritional growth factors and is, thus, unlikely to directly influence the risk of cancer [5]. In fact, the mechanisms by which adults attain body height increase pancreatic cancer risk have not been clearly identified. Greater adult body height may be related to increased exposure to insulin-like growth factor 1, which may influence cancer development by inhibiting apoptosis and stimulating proliferation, adhesion, and cell migration. In addition, taller people have more cells and, thus, there is a greater likelihood for mutations leading to cancer development [36,37]. Associations of red and processed meat intake with the risk of PC are controversial in the literature, with inverse, direct, and non-significant associations and without evident disparities by sex [38]. It is of interest to note that only women in our study appear to be at risk despite much lower meat intake compared to men.

Our findings of inverse associations between **breastfeeding duration** and PC risk are consistent with the results of two independent Norwegian cohorts. The more recent study comprising 588 incident cases reported a HR of 0.37 for those who had accumulated an overall duration of breastfeeding of more than 25 months [14]. Based on a different sample, a possible inverse association was suggested between breastfeeding and PC risk, with an RR of 0.87 per 12-month increment [39]. Results from these two Norwegian cohorts differ from others that do not find an association of breastfeeding and PC risk using a dichotomous variable for breastfeeding, i.e., ever/never breastfed [15,24,40,41,42,43], in line with our findings when applying this dichotomous approach. According to the literature, only studies considering longer periods of cumulative breastfeeding appear to have observed statistically significant associations. Furthermore, in the present study within EPIC, only the duration of breastfeeding of more than six months was significantly inversely associated with PC risk. Thus, it is likely that a potentially protective effect of breastfeeding only manifests itself once a certain threshold has been crossed. This should be further analysed in a pooled analysis to ensure sufficient statistical power.

The same inverse association might hold for a possible dose–response relationship between the number of FTP and PC risk in our results, despite not being statistically significant. Such a relationship had been suggested in a systematic review by Zhu et al. [23] for two pregnancies compared to nulliparity (RR 0.86, 9% CI 0.80–0.93) but not by Guan et al. [19] who reported a non-significant association when comparing the highest with the lowest parity group (RR 0.86, 95% CI 0.73–1.02).

The finding of a protective effect of **prolonged HRT** use in postmenopausal women has been observed in previous studies. Both ever compared to never and cumulative duration of HRT use were associated with PC risk in a prospective cohort from 2017 [16] and in a case–control study from 2020 [17]. Additionally, an investigation in the Pancreatic Cancer Case-Control Consortium (PanC4) from 2016 observed a statistically significant association, but only in women with hysterectomy [18], and investigations in The Malmö Diet and Cancer Study in 2018 suggested that the protective effect is restricted to oestrogen-only formulations of HRT [15]. Interestingly, a meta-analysis from 2015—not including the studies listed above—concluded that there was no significant association between HRT use and PC [22]. The authors, however, focused on ever use of HRT, rather than on the cumulative duration of its use. A recent cohort-based meta-analysis of 2023 stratified by HRT formulations showed inverse associations with risk of PC for women receiving estrogen-only and estrogen plus progestin HRT [44]. Most studies—including our own—were not able to analyse the effect of specific types of HRT. This proves difficult to generalise results across time and geographical regions as prescribing patterns and pharmaceutical formulations vary globally and over time [45,46,47].

Our findings of lower PC risk for prolonged breastfeeding and prolonged use of HRT can be explained by an underlying **biological mechanism** according to which oestrogen exposure generally protects against developing PC [37,48]. The oestrogen receptors Erα and Erβ have been shown to be expressed in the exocrine tissue of the pancreas and might play a role in tumour growth, either inhibiting tumour growth directly, as suggested in vitro [10] and in research on rat models [12], or via a moderating effect similar to what has been shown for the carcinogenesis of breast, colorectal and ovarian cancers [48,49,50,51]. In colorectal cancer, for example, the relationship of oestrogen receptor expression and tumour growth is likely to be the underlying biological mechanism explaining an observed risk reduction associated with HRT use [46]. The role of endogenous oestrogen in breast cancer risk and prognosis, on the other hand, is paradoxical, as it either confers a greater (prospective epidemiological studies) or a lower risk (clinical trials), depending on the study type [52]. The exact role of oestrogen receptors in pancreatic tumour growths, however, is not yet fully understood and is subject to ongoing research [11,13,53,54,55]. Intriguingly, it has recently been reported that knocking-out oestrogen-related receptor γ in acinar cells in mice results in acinar-ductal metaplasia, an early event in pancreatic carcinogenesis [56]. More comprehensive epidemiological studies would be required to develop more detailed hypotheses about a possible involvement of specific sex hormones in pancreatic carcinogenesis.

Besides a direct hormonal influence of breastfeeding on the development of PC via elevated oestrogen exposure, the observed inverse association may result from an indirect effect of breastfeeding on glucose metabolism and insulin sensitivity. It has been shown that each additional six months of breastfeeding reduces diabetes risk by 27% in the EPIC-Potsdam study; a subsequent meta-analysis resulted in a risk reduction of 20% [57]. Since a history of diabetes considerably increases the risk of PC [4], diabetes might constitute a mediating factor such that its effect on PC is reduced by breastfeeding, which, thus, indirectly lowers the risk of PC. This explanation would also be consistent with our finding of a threshold of around six months of breastfeeding, which must be crossed before a significant protective effect can be observed.

Given the profound physiological changes associated with pregnancy and breastfeeding, it is difficult to determine any single biological mechanism underlying a potential association between breastfeeding and pancreatic carcinogenesis. It could also be that abstaining from known risk factors (e.g., alcohol consumption, smoking) during pregnancy and periods of breastfeeding could result in the observed inverse risk estimates and is, thus, utterly independent of biological mechanisms related to breastfeeding.

### Strengths and Limitations

With 717 PC cases, this is one of the largest cohort studies examining the relationship between female reproductive and hormonal factors and the risk of PC. Owing to the prospective design, this study is less prone to observer bias and can establish a possible temporal relationship between exposures and outcomes. Also, unlike other studies, we have been able to examine a wide range of reproductive and hormonal factors and take into account the most relevant confounders.

The study carries inherent limitations of self-reported information on early reproductive history (e.g., age at menarche, use of OCs). However, since misclassification is independent of outcome, it would be non-differential but could have resulted in underestimated risk associations. While the majority of the EPIC-cohort represents the general population, some participants were recruited via blood donation, screening, or occupational health insurance programmes, introducing heterogeneity between study centres and limiting the generalisability of results. In order to reduce selection and measurement bias due to varying recruitment and research procedures in different study centres, the analyses were stratified by study centre. This stratification also contributes to tackling potential residual confounding due to regional differences regarding, for instance, healthy lifestyle choices and diet. There is no data collected on ethnicity in EPIC; not adjusting for this factor could have resulted in residual confounding [3].

A major limitation of this study is the lack of information on the exact pharmaceutical formulations of OC and HRT drugs taken by the participants. Data on the use of types of exogenous hormones would allow a much more precise evaluation of exposures, reducing residual confounding and also facilitating comparison across populations with different prescribing practices. Preventive interventions such as HRT are more likely to be chosen by individuals with a larger social capital and greater health literacy, who are healthier than the general population. This constitutes a potential source of residual confounding, which we sought to reduce in the present study by adjusting for other factors that are indicative of health choices, such as smoking status, BMI, and schooling with no major effect on risk estimates. Further, the cumulative exposure of established risk factors will be lower the more time a woman spends pregnant and during years of breastfeeding, which in turn might be an explanation for observed inverse associations; that we, however, could not explore. We took great care in limiting the likelihood of overestimating risk estimates by carefully choosing the optimal reference category, i.e., including only affected women while modelling quantitative reproductive and hormonal factors, such as parous women in the reference category for age at first pregnancy or breastfeeding women in the reference category for cumulative duration of breast feeding.

## 5. Conclusions

The incidence of PC is increasing in high-income countries and shows evident sex disparities. Based on a large prospective cohort, we confirmed a higher risk for men that could not be explained by the presence/absence of established risk factors compared to women. Nevertheless, we suspect differences in risk profiles by sex, and further research into sex disparities is greatly needed. We further identified prolonged time of breastfeeding and of HRT use as potential protective factors associated with PC risk. Pooled analyses of cohort studies, distinct analyses by menopausal status, more comprehensive information on the intensity and type of HRT, and assessment of lifestyle changes during pregnancy and breastfeeding are required to confirm and enrich these findings and make them relevant for targeted health interventions.

## Figures and Tables

**Figure 1 cancers-17-02275-f001:**
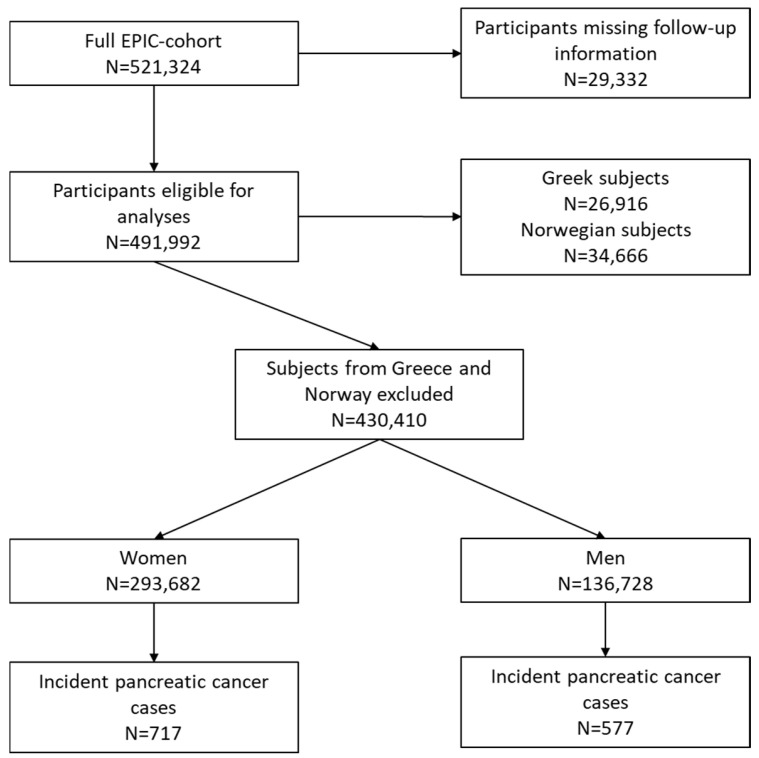
Flow diagram of the cohort, showing the numbers of included and excluded individuals at different stages and the number of incident pancreatic cancer cases in EPIC.

**Table 1 cancers-17-02275-t001:** Characteristics of the EPIC-cohort at baseline by sex (excluding Greece and Norway). Unless stated otherwise, data are given in the form of counts (column %) or median (IQR).

	FEMALE		MALE	
**Characteristics**	**Incident PC** n = 717	**Non-cases** n = 292,965	**Incident PC** n = 577	**Non-cases** n = 136,151
length of follow up (years)	10 (6, 13)	15 (14, 16)	10 (6, 13)	16 (12, 17)
age at diagnosis (years)	68 (62, 73)	-	67 (62, 72)	-
age at recruitment (years)	58 (52, 62)	51 (45, 57)	58 (52, 62)	53 (46, 59)
BMI (kg/m^2^)	25.0 (23, 28)	24.0 (22, 27)	26.3 (24, 29)	26.0 (24, 28)
alcohol intake at recruitment	*Missing 2%*	*Missing 1%*	*Missing 2%*	*Missing 3%*
non drinker	112 (16)	43,513 (15)	38 (7)	8417 (6)
>0–3 w/>0–6 m	208 (30)	86,742 (30)	114 (20)	33,753 (25)
>3–12 w/>6–12 m	198 (28)	87,585 (30)	86 (15)	22,342 (16)
>12–24	103 (15)	43,230 (15)	126 (22)	27,329 (20)
>24–60	71 (10)	26,429 (9)	138 (24)	32,730 (24)
≥60 w/>60–96 m	10 (1)	2274 (1)	47(8)	7238 (5)
≥96 m	-	-	13 (2)	1729 (1)
*continuous (g/d) (0.01% miss.)*	*3.6 (0.6, 11.9)*	*4.1 (0.6, 12)*	*15.8 (5.2, 34.5)*	*12.5 (3.9, 29.5)*
meat intake (g/d)	88 (59, 126)	86 (53. 122)	133 (90, 176)	121 (80, 167)
type 2 diabetes	*Missing 13%*	*Missing 8%*	*Missing 10%*	*Missing 12%*
no	599 (83)	264,153 (89)	472 (82)	115,572 (85)
yes	32 (4)	6208 (2)	36 (6)	4486 (3)
unknown	3	1097 (1)	12(2)	2
smoking status				
never	361 (50)	167,470 (57)	136 (24)	45,457 (33)
former	151 (21)	66,229 (23)	213 (37)	49,891 (37)
current	193 (27)	53,257 (18)	220 (38)	38,737 (28)
unknown	12 (2)	6009 (2)	8 (1)	2066 (2)
smoking history				
never	361 (50)	167,470 (57)	136 (24)	45,457 (33)
current, 1–15 cig/day	120 (17)	35,915 (12)	137 (24)	24,282 (18)
current, 16–25 cig/day	58 (8)	14,538 (5)	59 (10)	10,708 (8)
current, 26+ cig/day	15 (2)	2804 (1)	24 (4)	3747 (3)
former, quit ≤ 10 years	64 (10)	26,168 (9)	90 (16)	19,006 (14)
former, quit 11–20 years	28 (4)	20,901 (7)	50 (9)	15,385 (11)
former, quit 20+ years	59 (8)	19,160 (6)	73 (13)	15,500 (11)
unknown	12 (2)	6009 (2)	8 (1)	2066 (2)
**FEMALE Characteristics**			**Incident PC** n = 717	**Non-cases** n = 292,965
** *Reproductive Factors—menarche* **				
**age at menarche** (years), (4% missing)				
never			-	43 (0.01)
≤9			2 (0.3)	1647 (0.5)
10–12			243 (31)	113,605 (35)
13–15			418 (54)	181,664 (55)
≥16			74 (10)	18,818 (6)
**duration menstrual cycles** (years) (15% excluding participants with 0 and missing age at Menarche)			35 (31,38)	33 (28, 36)
**duration menstrual cycles without OC** (years) (15% missing excluding participants with 0 and missing age at Menarche and among pill non-users)			32 (26, 36)	29 (22, 34)
** *Reproductive Factors—pregnancy* **				
**full term pregnancy** (5% missing)				
no			88 (12)	45,148 (15)
yes			602 (84)	232,795 (81)
**number of full-term pregnancies,** (7% missing)			2 (1, 3)	2 (1, 3)
**age at first full-term pregnancy** (years) (0.4% missing among child bearing women)			24 (22, 28)	24 (22, 27)
**breastfeeding** (11% missing)				
no			160 (22)	77,799 (25)
yes			480 (70)	178,626 (63)
**duration breast feeding among ever fed** (months), (1% missing)			6 (3, 12)	6 (3, 11)
** *Reproductive Factors—menopause* **				
**menopausal status**				
premenopausal			68 (10)	101,184 (34)
postmenopausal			524 (73)	130,272 (43)
perimenopausal/unknown			102 (14)	52,532 (20)
surgical menopause			23 (3)	8977 (3)
**age at menopause** (years) (excluding premenopausal women, 47% missing)			50 (47, 52)	50 (46, 52)
** *Hormonal Factors—OC and HRT* **				
**use of OC** (3% missing)				
no			352 (49)	112,097 (38)
yes			336 (47)	171,435 (59)
**duration of OC use** (years), (47% missing) (10% missing among ever Pill users)			6 (2, 11)	6 (2,10)
**use of HRT** (6% missing: 11% cases, 7% non-cases)				
no			424 (59)	198,837 (68)
yes			212 (30)	71,400 (25)
**use of HRT among postmenopausal women** (8% missing: 11% cases, 6% non-cases)				
no			294 (63)	70,626 (58)
yes			175 (37)	51,674 (36)
**duration of HRT use** (years), (77% missing, 19% missing among ever hormone users)			2.0 (1, 6)	2.4 (1, 5)
**duration of HRT use among all hormone users** (years), (10% missing among ever hormone users)			2.5 (1, 7)	2.5 (1, 5)
**duration of HRT use among postmenopausal women** (years), (61% missing)			2 (1, 7)	3 (1, 6)
** *Hormonal Factors—surgery* **				
**hysterectomy** (11% missing)				
no			495 (69)	231,889 (78)
yes			99 (14)	32,734 (11)
**Ovariectomy** (20% missing)				
no			520 (67)	238,934 (73)
yes			57 (8)	20,266 (7)

**Table 2 cancers-17-02275-t002:** Multivariable adjusted * Hazard Ratios and 95% CI overall and in females and males for established risk factors in the EPIC cohort (n = 430,410).

FACTORS	All Subjects n = 430,410 1294 Cases	FEMALE n = 293,682 717 Cases	MALE n = 136,728 577 Cases
**Age**	**1.08 (1.07–1.09)**	**1.09 (1.08–1.10)**	**1.08 (1.06–1.09)**
**Sex** age and center adjusted	-	Reference	**1.39 (1.24–1.57)**
**Sex** fully adjusted	-	Reference	**1.31 (1.15–1.49)**
**Diabetes**	**1.74 (1.35–2.23)**	**1.74 (1.21–2.51)**	**1.72 (1.21–2.43)**
**BMI [kg/m^2^]**			
<25	Reference	Reference	Reference
25–30	0.99 (0.87–1.12)	0.99 (0.84–1.17)	0.97 (0.81–1.18)
≥30	1.09 (0.92–1.29)	0.98 (0.78–1.24)	1.22 (0.95–1.57)
*per 5 unit increment*	1.07 (0.99–1.14)	1.05 (0.96–1.14)	1.10 (0.98–1.24)
**Body height [cm]**/per 5 unit incr.	**1.07 (1.03–1.12)**	**1.12 (1.06–1.20)**	1.03 (0.96–1.09)
**Highest school level**			
None	Reference	Reference	Reference
Primary school completed	1.12 (0.77–1.63)	1.21 (0.71–2.05)	0.98 (0.57–1.68)
Technical/prof. school	1.02 (0.69–1.51)	1.08 (0.62–1.88)	0.93 (0.53–1.63)
Secondary school	1.00 (0.67–1.51)	0.97 (0.55–1.72)	1.04 (0.58–1.87)
Longer education	1.02 (0.68–1.51)	1.04 (0.59–1.83)	0.92 (0.52–1.61)
**Smoking status**			
Never	Reference	Reference	Reference
Former	1.05 (0.91–1.21)	0.94 (0.77–1.15)	1.21 (0.97–1.51)
Current	**1.75 (1.52–2.01)**	**1.74 (1.45–2.10)**	**1.83 (1.46–2.29)**
**Cigarettes smoked/day**			
1–10	Reference	Reference	Reference
11–20	**1.59 (1.33–1.91)**	**1.74 (1.35–2.24)**	**1.49 (1.16–1.91)**
21–30	**1.95 (1.45–2.61)**	**1.91 (1.18–3.10)**	**1.91 (1.32–2.76)**
>30	1.17 (0.60–2.27)	1.52 (0.48–4.79)	1.01 (0.44–2.29)
*continuous, 10/d*	**1.28 (1.07–1.49)**	**1.42 (1.27–1.59)**	**1.21 (1.10–1.32)**
**Duration of smoking**			
0.08–10 years	Reference	Reference	Reference
11–20 years	1.04 (0.78–1.37)	0.94 (0.62–1.40)	1.13 (0.77–1.67)
21–30 years	**1.32 (1.02–1.70)**	1.26 (0.88–1.81)	1.37 (0.96–1.97)
31–40 years	**1.65 (1.29–2.09)**	**1.80 (1.29–2.53)**	**1.56 (1.11–2.22)**
41–50 years	**1.55 (1.18–2.03)**	**1.59 (1.06–2.40)**	**1.55 (1.06–2.25)**
>50 years	1.57 (0.85–2.91)	1.68 (0.59–4.76)	1.55 (0.71–3.35)
*continuous/10 years*	**1.16 (1.09–1.23)**	** *1.20 (1.10–1.31)* **	**1.13 (1.05–1.22)**
**Time since quitting smoking**			
0.08–10 years	Reference	Reference	Reference
11–20 years	**0.51 (0.40–0.65)**	**0.53 (0.34–0.84)**	**0.59 (0.43–0.79)**
21–30 years	**0.63 (0.49–0.80)**	0.72 (0.46–1.11)	**0.65 (0.47–0.89)**
31–40 years	**0.65 (0.47–0.91)**	0.72 (0.41–1.27)	0.69 (0.44–1.08)
>40 years	0.61 (0.32–1.17)	0.55 (0.19–1.61)	0.72 (0.31–1.69)
*continuous/10 years*	**0.82 (0.77–0.88)**	*0.89 (0.76–1.04)*	**0.85 (0.78–0.93)**
**Alcohol intake at recr.**			
non drinker	Reference	Reference	Reference
>0–6 (m)/>0–3 (w)	0.89 (0.72–1.09)	0.97 (0.76–1.25)	0.75 (0.51–1.09)
>6–12 (m)/>3–12 (w)	0.91 (0.73–1.12)	0.98 (0.76–1.26)	0.76 (0.52–1.13)
>12–24	1.02 (0.82–1.28)	1.08 (0.81–1.44)	0.87 (0.59–1.27)
>24	1.14 (0.91–1.42)	1.34 (0.98–1.83)	0.89 (0.62–1.28)
*continuous, 15 g/d*	**1.08 (1.03–1.13)**	**1.11 (1.01–1.21)**	** *1.06 (1.01–1.12)* **
**Meat intake at recr.**			
*≤65*	Reference	Reference	Reference
*66–102*	1.14 (0.96–1.36)	1.07 (0.87–1.32)	1.33 (0.95–1.87)
*103–144*	1.04 (0.87–1.26)	1.05 (0.84–1.32)	1.05 (0.74–1.48)
*≥145*	**1.35 (1.10–1.65)**	**1.44 (1.11–1.87)**	1.27 (0.89–1.79)
*continuous, 100 g/d*	**1.15 (1.03–1.27)**	**1.18 (1.02–1.36)**	1.07 (0.92–1.25)

* If applicable, adjusted for age at recruitment, sex, smoking history, BMI, alcohol intake at recruitment and self-reported diabetes at recruitment.

**Table 3 cancers-17-02275-t003:** Hazard Ratios for reproductive and hormonal factors and pancreatic cancer risk in women of the EPIC cohort (n = 293,682, pancreatic cancer events n = 717).

Factors	HRs (95% CI) Adjusted for Age, Stratified by Study Centre	HRs (95% CI) Adjusted for Age, Education, Body Height, BMI, Smoking, Alcohol Consumption, Type 2 Diabetes Status; Stratified by Study Centre
** *Reproductive Factors—menarche* **		
**Age at menarche (years)**		
<12 (98 cases)	reference	
12 (126 cases)	0.91 (0.69–1.18)	0.91 (0.69–1.19)
13 (151 cases)	0.82 (0.63–1.06)	0.83 (0.64–1.07)
14 (154 cases)	0.85 (0.66–1.10)	0.86 (0.66–1.12)
≥15 (154 cases)	0.95 (0.73–1.24)	0.96 (0.74–1.25)
*continuous (per age-year)*	*1.00 (0.96–1.05)*	*1.01 (0.96–1.06)*
≤12 (224 cases)	reference	
>12 (387 cases)	0.89 (0.76–1.06)	0.90 (0.76–1.07)
**Cumulative duration menstrual cycling (years)**		
0–27.10 (55 cases)	reference	
27.11–31.28 (83 cases)	0.97 (0.68–1.38)	1.00 (0.71–1.43)
31.29–34.19 (117 cases)	1.03 (0.73–1.43)	1.08 (0.77–1.51)
34.20–36.91 (146 cases)	0.99 (0.72–1.38)	1.03 (0.74–1.44)
36.92–54.0 (178 cases)	1.04 (0.76–1.43)	1.11 (0.81–1.54)
continuous (per year)	1.01 (0.99–1.02)	1.01 (0.99–1.03)
≤33.04 (198 cases)	reference	
>33.04 (381 cases)	1.09 (0.92–1.31)	1.13 (0.94–1.35)
**Cumulative duration menstrual cycling without OC (years)**		
≤20.24 (52 cases)	reference	
20.25–27.05 (82 cases)	0.99 (0.69–1.42)	1.03 (0.72–1.47)
27.06–31.50 (107 cases)	0.98 (0.69–1.38)	1.02 (0.72–1.45)
31.51–35.49 (137 cases)	0.98 (0.69–1.37)	1.01 (0.72–1.42)
>35.50 (164 cases)	0.99 (0.71–1.38)	1.06 (0.76–1.48)
*continuous (per year)*	*0.99 (0.98–1.01)*	*0.99 (0.98–1.01)*
≤29.48 (202 cases)	reference	
>29.48 (342 cases)	0.91 (0.75–1.09)	0.95 (0.79–1.14)
** *Reproductive Factors—pregnancy* **		
**Full-term pregnancy**		
no (97 cases)	reference	
yes (589 cases)	0.89 (0.71–1.10)	0.87 (0.70–1.09)
**Number of full-term pregnancies**		
0 (97 cases)	reference	
1 (119 cases)	0.94 (0.72–1.24)	0.91 (0.69–1.20)
2 (265 cases)	0.89 (0.70–1.13)	0.89 (0.70–1.13)
3 (133 cases)	0.85 (0.66–1.11)	0.84 (0.64–1.09)
≥4 (68 cases)	0.78 (0.57–1.08)	0.77 (0.56–1.06)
*continuous*	*0.96 (0.90–1.02)*	*0.96 (0.90–1.02)*
≤2 (472 cases)	reference	
>2 (201 cases)	0.89 (0.76–1.07)	0.89 (0.75–1.06)
**Age at first full-term pregnancy (years) among childbearing women**		
<22 (183 cases)	reference	
22–24 (113 cases)	0.92 (0.72–1.16)	0.94 (0.74–1.19)
25–29 (206 cases)	0.88 (0.72–1.08)	0.96 (0.78–1.18)
≥30 (84 cases)	0.95 (0.73–1.24)	1.04 (0.79–1.36)
*continuous (per age-year)*	*0.99 (0.98–1.02)*	*1.01 (0.98–1.03)*
≤24 (296 cases)	reference	
>24 (290 cases)	0.93 (0.79–1.10)	1.00 (0.84–1.19)
**Breastfeeding**		
no (160 cases)	reference	
yes (480 cases)	0.98 (0.82–1.17)	0.99 (0.83–1.19)
**Cumulative duration of breast feeding (months) among breast feeding women**		
≤2.07 (114 cases)	reference	
2.08–5.00 (109 cases)	0.84 (0.65–1.10)	0.89 (0.68–1.16)
5.01–8.50 (78 cases)	**0.69 (0.51–0.92)**	**0.73 (0.54–0.98)**
8.51–15.00 (95 cases)	**0.71 (0.54–0.93)**	0.76 (0.57–1.01)
>15.00 (82 cases)	**0.73 (0.54–0.98)**	0.78 (0.58–1.05)
*continuous (per month)*	*0.99 (0.98–1.00)*	*0.99 (0.98–1.01)*
≤5.74 (237 cases)	reference	
>5.74 (241 cases)	**0.74 (0.61–0.89)**	**0.77 (0.64–0.93)**
** *Reproductive Factors—menopause* **		
**Menopausal status**		
Pre (68 cases)	0.87 (0.61–1.22)	0.89 (0.63–1.26)
Peri/unknown (102 cases)	1.09 (0.85–1.39)	1.07 (0.83–1.36)
Post (524 cases)	reference	
Surgical (23 cases)	0.74 (0.48–1.12)	0.68 (0.44–1.05)
**Age at menopause (years)**		
<46 (109 cases)	reference	
46–50 (164 cases)	1.03 (0.81–1.31)	1.04 (0.82–1.33)
51–52 (74 cases)	0.96 (0.71–1.28)	0.97 (0.72–1.31)
≥53 (100 cases)	1.02 (0.77–1.34)	1.06 (0.80–1.39)
*continuous (per year)*	*0.99 (0.98–1.02)*	*1.00 (0.98–1.02)*
≤50 (273 cases)	reference	
>50 (174 cases)	0.97 (0.80–1.17)	0.99 (0.82–1.21)
** *Hormonal Factors—OC and HRT* **		
**Ever use OC**		
no (352 cases)	reference	
yes (336 cases)	1.16 (0.98–1.36)	1.13 (0.96–1.34)
**Cumulative duration of OC use (years) among OC users**		
≤2 (89 cases)	reference	
2–5 (49 cases)	0.84 (0.59–1.19)	0.86 (0.61–1.23)
6–10 (75 cases)	1.04 (0.76–1.42)	1.04 (0.76–1.43)
>10 (85 cases)	1.05 (0.78–1.43)	1.04 (0.76–1.41)
*continuous (per year)*	*1.01 (0.99–1.02)*	*1.01 (0.99–1.02)*
≤6 (147 cases)	reference	
>6 (151 cases)	1.20 (0.95–1.52)	1.19 (0.93–1.50)
**Ever use of HRT**		
no (424 cases)	reference	
yes (212 cases)	1.04 (0.87–1.23)	1.01 (0.85–1.21)
**Cumulative duration of HRT use (years) among HRT users**		
≤1 (74 cases)	reference	
1.1–2.5 (30 cases)	0.83 (0.54–1.28)	0.82 (0.53–1.27)
2.6–5.0 (38 cases)	0.69 (0.46–1.02)	0.69 (0.47–1.03)
>5.0 (50 cases)	**0.63 (0.43–0.914)**	**0.64 (0.44–0.93)**
*continuous (per year)*	*0.98 (0.96–1.02)*	*0.99 (0.96–1.02)*
≤2.42 (102 cases)	reference	
>2.42 (90 cases)	**0.71 (0.53–0.95)**	**0.72 (0.54–0.97)**
** *Hormonal Factors—surgery* **		
**Hysterectomy**		
no (495 cases)	reference	
yes (99 cases)	0.95 (0.76–1.18)	0.94 (0.75–1.17)
**Ovariectomy**		
no (520 cases)	reference	
yes (57 cases)	0.89 (0.68–2.17)	0.87 (0.65–1.15)

Only one reproductive factor is tested at a time. Missing: Full term pregnancy (31 cases), breastfeeding (77 cases), OC use (29 cases), HRT use(no cases), Duration of menstrual cycle (138 cases), Duration of menstrual cycle without OC (173 cases), Cumulative duration of breastfeeding (239 cases), Cumulative duration of OC Use (419 cases), Cumulative duration of HRT Use among HRT users (20 cases), Age at Menarche (34 cases), Age at first full term pregnancy (131 cases), Age at Menopause (270 cases), Number of full term pregnancies (44 cases), Ever use of HRT (81 cases), Hysterectomy (123 cases), Ovariectomy (140 cases).

## Data Availability

EPIC data and biospecimens are available for investigators who seek to answer important questions on health and disease in the context of research projects that are consistent with the legal and ethical standard practices of the International Agency for Research on Cancer (IARC), WHO, and the EPIC centres. The primary responsibility for accessing the data, obtained in the frame of the present publication, belongs to the EPIC centres that provided it. Access to EPIC data can be requested from the EPIC Steering Committee, as detailed in the EPIC-Europe Access Policy (https://epic.iarc.who.int/access/ (accessed on 1 June 2025)).

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
