# Peer review of "Sex Disparities and Female Reproductive and Hormonal Factors Associated with Risk of Pancreatic Cancer in the European Prospective Investigation into Cancer and Nutrition (EPIC) Cohort"

_cancers, 2025, doi:10.3390/cancers17142275_

Round 1

Reviewer 1 Report

Comments and Suggestions for Authors

Thank you for the opportunity to review the manuscript ID: cancers-3706962. This study aimed to analyze general and sex-related risk factors for pancreatic cancer in the European Prospective Investigation into Cancer and Nutrition (EPIC) cohort.

This manuscript is well clear, readable and informative for the topic it deals with.

In the Introduction section of this paper, the basis of the research question is well presented, that is, the gap in knowledge of differences in the incidence of pancreatic cancer according to gender is highlighted. In the Introduction section, as well as in other sections of this work, relevant, contemporary literature is cited.
Lines 130-133: State the goal of this manuscript in more detail and precision. The study population is specified in the following paragraphs, so it is not necessary to specify the size of the studied population in this sentence.

In the Methods section (as well as in the Appendix) Study design, Study population, Follow-up, Variables, Outcomes, Statistical analysis are described in detail.
Lines 136-137: Explain the number of women being more than twice than men in the EPIC study.
Lines 137-138: List the criteria for inclusion in the EPIC study.
Line 160: Indicate the date with which the incident cases of pancreatic cancer analyzed in this manuscript were identified in this study. 
Line 175: State whether the data presented in this paper were collected only at the time of inclusion of the participants in the EPIC study?
If data were collected only `at recruitment`, discuss the lack of data on changes in selected exposures as a limitation of this study.         

Table 1: Explain the differences in values ​​for the category `Never` for the variables `Smoking status` and `Smoking intensity`.

Table 2:
Unlike the descriptive data presented in Table 1 (for the variables `Smoking status` and `Smoking intensity`, which included the category `Never`, as well as the variable `alcohol intake at recruitment` which included the category `non drinker`), Multivariable adjusted Hazard Ratios on Table 2 for the variable `Smoking status` retains the category `Never`, and for the variable `alcohol intake at recruitment` retains the category `non drinker` as a reference category.
Explain why on Table 2 for the variables `Cigarettes smoked / day` and `Duration of smoking` the reference category was `≤10`?
Explain whether this means that the `≤10` category also includes never smokers?
Explain whether for the variables `Time since quitting smoking` the reference category `≤10 years` includes current smokers?

Table 3: Similar to the comments for Table 2, align all variables on Table 3 that show cumulative values ​​(for example, one way for `Full-term pregnancy` and `Number of full-term pregnancies`, and another way for `Breastfeeding` and `Cumulative duration of breast feeding (months)`, etc).

Common comment for Table 2 and Table 3: Could the presented selection of reference categories in the investigated variables affect the values ​​of Multivariable adjusted Hazard Ratios (on Table 2) and the values ​​of Hazard Ratios (on Table 3).
If this selection of reference categories could have influenced the results, whether the effect was overestimated or underestimated.
Discuss these issues in the Limitations section of this manuscript.

In the Discussion section, the results of this work are explained in a consistent and comprehensive way. The comparison of the results of this manuscript with the results of other studies was carried out in an appropriate manner. Possible interpretations for the significant findings of this paper are presented in an adequate manner.

The `Strengths and Limitations` paragraph correctly states and discusses the limitations of this manuscript.

The Conclusions section reflects the most important results of this work. 

Author Response

Response to Reviewer 1 Comments

1. Summary

Thank you very much for taking the time to review this manuscript. And thank you for your kind words and the supportive views on our manuscript sections. This is much appreciated. Please find the detailed responses below and the corresponding revisions/corrections highlighted/in track changes in the re-submitted files.

2. Questions for General Evaluation

Reviewer’s Evaluation

Response and Revisions

Does the introduction provide sufficient background and include all relevant references?

Yes

Are all the cited references relevant to the research?

Yes

Is the research design appropriate?

Yes

Are the methods adequately described?

Can be improved

We have taken care of improving our method description according to the comments provided, please find details below.

Are the results clearly presented?

Can be improved

We have modified parts of the results section and aligned those to the comments, see details below.

Are the conclusions supported by the results?

Yes

3. Point-by-point response to Comments and Suggestions for Authors

Comments 1: Lines 130-133: State the goal of this manuscript in more detail and precision. The study population is specified in the following paragraphs, so it is not necessary to specify the size of the studied population in this sentence.

Response 1: Thank you for pointing this out. We agree with this comment. Therefore, we have added our aims more clearly but kept the number of cases, as this is a strength of our research and nicely follows the line of thoughts (i.e. updated research in the same cohort). [Our aim was to investigate associations of established risk factors for pancreatic cancer by sex, followed by in-depths analyses of hormonal and reproductive factors and their associations with pancreatic cancer risk in women, including parity, breastfeeding, and use of hormones– page 3, lines 135-138.]

Comments 2: Lines 136-137: Explain the number of women being more than twice than men in the EPIC study.

Response 2: Agree. We have, accordingly, added information to clarify the higher percentages of women. [The study centers recruited mainly from the general adult population, but also from health insurances, health conscious populations, or blood donors. In France, Norway, Utrecht and Naples only women were recruited – page 3, lines 146-149.]

Comments 3: Lines 137-138: List the criteria for inclusion in the EPIC study.

Response 3: We have added further information, as suggested. [Participant eligibility within each cohort was based on geographic or administrative boundaries and on a predefined age range. – page 3, lines 149-150.]

Comments 4: Line 160: Indicate the date with which the incident cases of pancreatic cancer analyzed in this manuscript were identified in this study.

Response 4: The censoring date is already mentioned on page 4, line 175, after Figure 1.

Comments 5: Line 175: State whether the data presented in this paper were collected only at the time of inclusion of the participants in the EPIC study? If data were collected only `at recruitment`, discuss the lack of data on changes in selected exposures as a limitation of this study.

Response 5: The data collected at baseline had not been re-assessed uniformly across all centers during follow-up of the study. We have added this information in materials & methods. [Baseline characteristics of participants were not re-assessed across all centers during follow-up of the study – page number, paragraph, and line.] It is already mentioned in the discussion as a limitation, page 4, lines 15-160.

Comments 6: Table 1: Explain the differences in values ​​for the category `Never` for the variables `Smoking status` and `Smoking intensity`.

Response 6: Thank you for pointing this out. Within the EPIC study, several questions were asked to capture smoking habits. “Smoking status” is derived from one question, whereas “smoking intensity” has been build based on several questions. Thus, it is possible that a non-smoker indicated “smoking status” = never (Question: “do/did you smoke?”) but later in the questionnaire the participant reported to smoke occasionally, thus “smoke intensity” = current pipe/cigar/occas (=occasionally). To clarify the latter, we have spelled out occas=occasionally. We cross-checked all other numbers (no typing errors), and changed the word “smoking intensity” to “smoking history”, as “history” better reflects the groupings and is the word used in the text.

Comments 7: Table 2:

Unlike the descriptive data presented in Table 1 (for the variables `Smoking status` and `Smoking intensity`, which included the category `Never`, as well as the variable `alcohol intake at recruitment` which included the category `non drinker`), Multivariable adjusted Hazard Ratios on Table 2 for the variable `Smoking status` retains the category `Never`, and for the variable `alcohol intake at recruitment` retains the category `non drinker` as a reference category.

Explain why on Table 2 for the variables `Cigarettes smoked / day` and `Duration of smoking` the reference category was `≤10`?

Explain whether this means that the `≤10` category also includes never smokers?

Explain whether for the variables `Time since quitting smoking` the reference category `≤10 years` includes current smokers?

Response 7: Thank you again for pointing this out. In Table 2, the reference categories for “Cigarettes smoked”, “Duration of smoking” and “Time since quitting smoking” retain never smokers AND those of unknown smoking status. We have now revised the reference category and excluded the never smokers and those with an unknown smoking status. The reference categories are for “Cigarettes smoked” 1-10, for “Duration of smoking” 0.08-10, and for “Time since quitting smoking” 0.08-10 (0.08 years = one month). We have modified the respective lines in Table 2.

The risk estimates slightly changed in effect size but remained unchanged in significance, as can be seen in the modified Table 2. We have, accordingly, changed the HR and 95%CI in the text. [abstract line 97, results line 260, discussion line 313]

Comments 8: Table 3: Similar to the comments for Table 2, align all variables on Table 3 that show cumulative values (for example, one way for `Full-term pregnancy` and `Number of full-term pregnancies`, and another way for `Breastfeeding` and `Cumulative duration of breast feeding (months)`, etc).

Response 8: Agree. We have, accordingly, modified Table 3 to emphasize this point. [Age at first full-term pregnancy (years) among childbearing women, Cumulative duration of breast feeding (months) among breast feeding women, Cumulative duration of OC use (years) among OC users]. The other variables already included lower cut-offs for the reference category.

Comments 9: Common comment for Table 2 and Table 3: Could the presented selection of reference categories in the investigated variables affect the values of Multivariable adjusted Hazard Ratios (on Table 2) and the values of Hazard Ratios (on Table 3).

If this selection of reference categories could have influenced the results, whether the effect was overestimated or underestimated.

Discuss these issues in the Limitations section of this manuscript

Response 9: Indeed, retaining for example never smokers in the reference categories for smoking variables resulted in slightly different HR in Table 2 compared to excluding never smokers from the reference category. However, associations were not uniformly greater or lower in effect size, this varied by variable and by sex. For cigarettes smoked, and as expected, retaining never smokers resulted in higher HR than omitting never smokers from the reference category. For duration of smoking, HR increased in women but decreased in men comparing omitted to retained never smokers in the reference category.

In principle, choosing the reference category does have an influence on HR. Therefore, we carefully chose the reference categories in hormonal and reproductive factors, such as “cumulative duration of breastfeeding” only among breastfeeding women, the non-breastfeeding women were omitted from the reference category.

HR for quantitative reproductive and hormonal factors would be overestimated if the reference category would include non-affected women. Such as nulliparous women in the reference category for “age at first full-term pregnancy” or nulliparous and/or non-breast-feeding but childbearing women in the reference category for “cumulative duration of breast feeding”. We have added this issue in the limitation section of the manuscript. [We took great care in limiting the likelihood of overestimating risk estimates by carefully choosing the optimal reference category, i.e. including only affected women while modelling quantitative reproductive and hormonal factors, such as parous women in the reference category for age at first pregnancy or breastfeeding women in the reference category for cumulative duration of breast feeding – page 14, lines 434-438.]

4. Response to Comments on the Quality of English Language

Point 1: The English is fine and does not require any improvement.

Response 1: Thank you very much

Reviewer 2 Report

Comments and Suggestions for Authors

General comments.

The article presented for the review is an analysis of data obtained from a multicenter high quality prospective analytic cohort study on the risk of pancreas cancer. Specifically, the analysis is devoted to factors related to sex disparities in the disease. As the risk has been found to be lower in women, a set of hormonal and reproductive factors was examined after adjustment for well-established other risk factors such as alcohol, smoking, type II diabetes, and BMI.  The leading hypothesis of this analysis is related to the possible protective role of oestrogens.

As data for this analysis are coming from a very large cohort and number of cases is sufficiently high, it provides a good basis for the statistical confidence (a strength of this analysis). However, the results of the analysis in most cases with few exceptions find very weak statistically unsignificant associations. Positive findings are discussed with conclusion about the necessity of further studies.

The general methodological, scientific and analytic level of the article is very good, and only some small non-essential remarks and questions will follow, and they are mainly related to the inherited and well-known disadvantages of observational epidemiological studies like.

  Special comments.

Lines 177-179.  This paragraph is related to the previous sub-section “Follow-up for cancer incidence… “.

Table 1.  Alcohol intake at recruitment – “>0-3w / >0-6m” can be misinterpreted. Mathematically correct writing would be “0<x≤3”

Line 297. Instead of “height”, it is better to speak of “body height” (this is everywhere). For this interesting finding, is there any evidence, that “genetic, environmental, hormonal, and nutritional growth factors” influence body height, and that these factors are related to cancerogenesis? References?

Lines 313-317. What could be the biological hormonal explanation for the protective association with prolonged breastfeeding? Hypothesis, if there are no studies? Sorry: lines 369-374 better to move to previous lines.

Lines 383-384.  Non-differential misclassification is shifting the result in the direction of 1 (null hypothesis). Does it mean, that the result obtained theoretically could be even more pronounced?

Was any attention paid to possible interactions?

Certainly not all the counfounders explaining for residual confounding could be accounted for. For me it would be interesting to now only diabetes, but other comorbidities as well. The male sex is associated with an increased risk of several gastrointestinal diseases. What about sex-related eating habits?  

Author Response

Response to Reviewer 2 Comments

1. Summary

Thank you very much for taking the time to review this manuscript. And thank you for your kind and positive feedback. Please find the detailed responses below and the corresponding revisions/corrections highlighted/in track changes in the re-submitted files.

2. Questions for General Evaluation

Reviewer’s Evaluation

Response and Revisions

Does the introduction provide sufficient background and include all relevant references?

Yes

Are all the cited references relevant to the research?

Yes

Is the research design appropriate?

Yes

Are the methods adequately described?

Yes

Are the results clearly presented?

Yes

Are the conclusions supported by the results?

Yes

3. Point-by-point response to Comments and Suggestions for Authors

Comments 1: Lines 177-179. This paragraph is related to the previous sub-section “Follow-up for cancer incidence… “.

Response 1: Thank you for pointing this out. We agree with this comment. Therefore, we have moved the paragraph to the previous sub-section, as suggested (page 5, lines 186-188).

Comments 2: Table 1. Alcohol intake at recruitment – “>0-3w / >0-6m” can be misinterpreted. Mathematically correct writing would be “0<x≤3”

Response 2: Agree. We have, accordingly, modified the categories to prevent misinterpretation. It now states 0-<3 or ≥60 (see table 1).

Comments 3: Line 297. Instead of “height”, it is better to speak of “body height” (this is everywhere). For this interesting finding, is there any evidence, that “genetic, environmental, hormonal, and nutritional growth factors” influence body height, and that these factors are related to cancerogenesis? References?

Response 3: True. We have, accordingly, added “body” whenever we speak of “height”, which indeed is “body height”. You may find this, for example, in the abstract on page 2, line 96.

Indeed, specific mechanisms that link greater adult body height with an increased risk of pancreatic cancer have not been clearly identified. But greater adult height may be related to increased exposure to endocrine and metabolic patterns. In addition, taller people have more cells and, thus, there is greater likelihood for mutations and cancer development. We have added an explanatory sentence and two further references. [In fact, the mechanisms by which adult attained body height increases pancreatic cancer risk have not been clearly identified. Greater adult body height may be related to increased exposure to insulin-like growth factor 1, which may influence cancer development by inhibiting apoptosis and by stimulation of proliferation, adhesion, and cell migration. In addition, taller people have more cells and, thus, there is greater likelihood for mutations leading to cancer development. – page 12, lines 321-327.]

Comments 4: Lines 313-317. What could be the biological hormonal explanation for the protective association with prolonged breastfeeding? Hypothesis, if there are no studies? Sorry: lines 369-374 better to move to previous lines.

Response 4: Already in the manuscript in the subsection biological plausibility we elaborate on potential mechanisms: “…the observed inverse association may result from an indirect effect of breastfeeding on glucose metabolism and insulin sensitivity…” Diabetes might be a mediating factor in the associations of breastfeeding with risk of pancreatic cancer such that the effect of diabetes on PC risk is reduced by breastfeeding and, thus, breastfeeding indirectly lowers risk of PC [page 13, line 388]. We would prefer to keep the biological plausibility in one block after the comparisons of our results with the available literature, according to the STROBE guidelines. Further, the common basis of the biological plausibility of HRT and breastfeeding is oestrogen related, thus this section nicely fits into the current flow of the manuscript.

Comments 5: Lines 383-384. Non-differential misclassification is shifting the result in the direction of 1 (null hypothesis). Does it mean, that the result obtained theoretically could be even more pronounced?

Response 5: Indeed, this could be true. Thank you for pointing this out. We have, accordingly, extended the existing sentence on misclassification in the discussion to emphasize this point. [… but could have resulted in underestimated risk associations – page 14, lines 412-413.]

Comments 6: Was any attention paid to possible interactions?

Response 6: We have performed interaction analyses for full-term pregnancies x breast feeding and HRT use x menopause. However, interaction terms were not statistically significant and were, therefore, omitted from analyses. We had not mentioned this before but now added a sentence in the methods section. [Interaction terms between pregnancies and breastfeeding as well as menopausal status and HRT use proved also to be non-significant and were, therefore, not included in the models – page 5, lines 218-220.]

Comments 7: Certainly not all the counfounders explaining for residual confounding could be accounted for. For me it would be interesting to now only diabetes, but other comorbidities as well. The male sex is associated with an increased risk of several gastrointestinal diseases. What about sex-related eating habits?

Response 7: That is an interesting point. We have only incomplete information on gastrointestinal diseases in EPIC at time of recruitment and could, therefore, not use this information. However, we do have diet available that already has been published multiple times in EPIC [e.g. meat intake PMID: 22610753]. According to the latest literature, there is no strong evidence that diet is associated with risk of pancreatic cancer [Diet, nutrition, physical activity and pancreatic cancer]. As there is limited but suggestive evidence available for red and processed meat, and to acknowledge your comment, we now added analyses on meat consumption. And observed a direct association of meat intake with PC risk, confined to women. Further adjusting for meat intake in our main analyses did not alter observed risk associations. We have added a line in table 2 and some text in the abstract, result and discussion sections. [Discussion: Associations of red and processed meat intake with risk of PC are controversial in the literature, with inverse, direct and non-significant associations and without evident disparities by sex. It is of interest to note that women in our study appear to be at risk despite much lower meat intake than men.– page 12, lines 327-330.]

4. Response to Comments on the Quality of English Language

Point 1: The English is fine and does not require any improvement.

Response 1: Thank you!

Round 2

Reviewer 1 Report

Comments and Suggestions for Authors

Thank you for the opportunity to re-review the manuscript ID: cancers-3706962.
The authors addressed most of my comments and corrected this paper in an appropriate way. I thank the authors.

However, several issuess remain, including:

1. Inconsistency of what is stated in the Methods section (on Lines 210-212) with what is presented in the Results section on Table 1.

Therefore, it is mandatory in the Methods section to specify in detail the definition applied in this study for the variables `Smoking status` and `Smoking intensity` (that is, `Smoking history` variable according to the revised version of this manuscript). 
Harmonize the definitions of the variables (and their categories) in the Methods section with the presentation of those variables in the Results section.

As for the attached response, such circumstances can and should be resolved either during the planning of the study, or during the realization of the study itself, during the formatting of the database, or during checking of the database (including cross-checking all numbers, etc.), etc.

2. Table 2: In the Methods section, provide a description of the variables `Cigarettes smoked / day` and `Duration of smoking`, with all the associated categories and an indication of whether they include nonsmokers.

3. Suggestion: Check and correctly enter the mentioned references (Lines 695-699) in the List of references.
